Combination of serum CST1 and HE4 for early diagnosis of endometrial cancer

Zhong Wenhui 1 2
Liu Yunliang 3
Zhang Liangming 2 4
Zhuang Wanzhen 2 5
Chen Jianlin 2 5
Huang Zhixin 2 6
Zheng Yue 2 5
Huang Yi hyi8070@126.com 2 5 7 8
1 Department of Clinical Laboratory, Fujian Maternity and Child Health Hospital, College of Clinical Medicine for Obstetrics & Gynecology and Pediatrics, Fujian Medical University , Fuzhou , People’s Republic of China
2 Shengli Clinical Medical College, Fujian Medical University , Fuzhou , People’s Republic of China
3 Department of Otolaryngology, Fujian Maternity and Child Health Hospital, College of Clinical Medicine for Obstetrics & Gynecology and Pediatrics, Fujian Medical University , Fuzhou , People’s Republic of China
4 Department of Clinical Laboratory, Fujian Provincial Hospital South Branch , Fuzhou , People’s Republic of China
5 Central Laboratory, Center for Experimental Research in Clinical Medicine, Fujian Provincial Hospital , Fuzhou , People’s Republic of China
6 Integrated Chinese and Western Medicine College, Fujian University of Traditional Chinese Medicine , Fuzhou , People’s Republic of China
7 Department of Clinical Laboratory, Fujian Provincial Hospital , Fuzhou , People’s Republic of China
8 Fujian Provincial Key Laboratory of Cardiovascular Disease, Fujian Provincial Key Laboratory of Critical Care Medicine , Fuzhou , People’s Republic of China
Uversky Vladimir
Electronic publication date: 2023 Dec 5
Publication date: 2023
Volume: 11
Electronic Location ID: e16424
Received 2023 Jul 26; Accepted 2023 Oct 18
Copyright: ©2023 Zhong et al.
Copyright year: 2023
Copyright holder: Zhong et al.
License: This is an open access article distributed under the terms of the Creative Commons Attribution License, which permits unrestricted use, distribution, reproduction and adaptation in any medium and for any purpose provided that it is properly attributed. For attribution, the original author(s), title, publication source (PeerJ) and either DOI or URL of the article must be cited.
License URL: https://creativecommons.org/licenses/by/4.0/

Keywords: CST1, HE4, Endometrial cancer (EC), Early diagnosis, Combined detection

Funding: Medical Vertical Project of Fujian Province 2020CXB001 Joint Fund of Science and Technology Innovation of Fujian Province 2021Y9024 Key Project of Natural Science Foundation of Fujian Province 2022J02048 Project of Natural Science Foundation of Fujian Province 2023J011222 This study was supported by the Medical Vertical Project of Fujian Province (Grant No. 2020CXB001) to Yi Huang, the Joint Fund of Science and Technology Innovation of Fujian Province (Grant No. 2021Y9024) to Yi Huang, the Key Project of Natural Science Foundation of Fujian Province (Grant No. 2022J02048) to Yi Huang, and the Project of Natural Science Foundation of Fujian Province (Grant No. 2023J011222) to Wenhui Zhong. The funders had no role in study design, data collection and analysis, decision to publish, or preparation of the manuscript.

==============================
Purpose

Optimal serological biomarkers have been absent for the early diagnosis of endometrial cancer, to date. In this study, we aimed to define the diagnostic performances of individual and combined detection of serum cysteine protease inhibitor 1 (CST1) with traditional tumor markers, including glycated antigen 125 (CA125) and human epididymis protein 4 (HE4), in patients with early-stage endometrial cancer (EC).

Methods

The performances of individual and combined detection of serum CST1, HE4, and CA125 were evaluated by enzyme-linked immunosorbent assay (ELISA) and chemiluminescent immunoassay, respectively. A training data set of 67 patients with early EC, 67 patients with endometrial benign lesion (EBL), and 67 healthy controls (HC) was used to develop a predictive model for early EC diagnosis, which was validated by an independent validation data set.

Results

In the training data set, serum CST1 and HE4 levels in the early EC group were significantly higher than in EBL/HC groups (P < 0.05), while there was no significant difference of serum CA125 level between the early EC and EBL/HC groups (P > 0.05). Serum CST1 and HE4 exhibited areas under the curve (AUC) of 0.715 with 31.3% sensitivity at 90.3% specificity, and 0.706 with 23.9% sensitivity at 95.5% specificity, respectively. Combined detection of serum CST1 and HE4 exhibited an AUC of 0.788 with 49.3% sensitivity at 92.5% specificity. The combination of serum CST1 and HE4 showed promise in diagnosis.

Conclusion

CST1 is a prospective serological biomarker for early EC diagnosis, and the combination of CST1 and HE4 contributes to the further improvement in the diagnosis of patients with early-stage EC.

Introduction

Endometrial cancer (EC) is an epithelial malignancy accounting for 20%–30% of malignant tumors of the female reproductive tract (Torre et al., 2015). In recent years, with the improvement of medical diagnosis and changes in human living conditions, the incidence of EC has been increasing and even spreading to younger age groups (Jemal et al., 2007; Siegel et al., 2021). Early diagnosis and timely treatment are essential for the prognosis of patients with EC, which currently has a 5-year survival rate of up to 95% in early EC patients while it is a dismal 17% in advanced EC patients (National Cancer Institute, 2015). Currently, histopathology is the gold standard for the diagnosis of EC; however, hysteroscopic biopsy and diagnostic scraping are not suitable for screening asymptomatic populations due to invasiveness and relatively complex operations. As a screening method, ultrasonography meets the barrier of ineffectively differentiating EC from precancerous lesions (Breijer et al., 2012). Serological biomarkers have the advantages of convenience and safety in screening for early EC. Unfortunately, current traditional tumor markers are absent for early diagnosis of EC, mainly owing to detection sensitivity and specificity issues. For example, glycated antigen 125 (CA125) and human epithelial protein 4 (HE4) for EC diagnosis are limited by interference from the female physiological cycle, and age and renal function, respectively (Buamah, 2000; Escudero et al., 2011; Nagy Jr et al., 2012). Therefore, developing new serological biomarkers is essential for the diagnosis of early EC.

The cystatin (CST) superfamily consists of endogenous or secreted proteins that maintain a balance with intracellular and extracellular cysteine proteases (Breznik et al., 2019); when this balance is broken, it may contribute to the occurrence of malignancies (Feldman et al., 2009). Recently, it has been reported that CSTs are intimately involved with malignancies, including lung cancer, breast cancer, and liver cancer, and responsible for the promotion of cancer cell proliferation, invasion and migration (Blanco et al., 2012; Choi et al., 2009; Dai et al., 2017; Jiang et al., 2015; Sousa-Pereira et al., 2014; Wang et al., 2021; Yoneda et al., 2009; Zhang et al., 2023). Additionally, serum cystatin SN (CST1) was also reported to be a valuable diagnostic biomarker for colorectal cancer and ESCC (Yoneda et al., 2009; Wang et al., 2021). Considering that the correlation between CST1 and EC has not yet been clarified, it is worth exploring whether there is ectopic high expression of CST1 in both cancerous tissues and sera of EC patients. Based on our pilot study, CST1 protein was shown to present aberrantly high expression in cancerous tissues of 37 patients with early EC, with a positive rate of 67.6% (25/37), which is remarkably higher than that in matched paracancerous tissues (P < 0.01). Additionally, by the detection of serum CST1 in a small cohort of 100 serum samples, including 30 cases with early EC at tumor node metastasis (TNM) I/II stage, 10 cases with advanced EC at TNM III/IV stage, 30 cases with EBL, and 30 HCs, we found that serum CST1 detection might be able to differentiate early EC from EBL and HC (P < 0.05, with an area under the curve (AUC) of up to 0.718 for early EC patients, with 43.3% sensitivity at 88.3% specificity; Fig. S1). Here, we aimed to comprehensively define the diagnostic performances of individual and combined detections of serum CST1 with traditional tumor markers (CA125 and HE4) for patients with early EC in a large cohort of 300 serum samples. A training set of 67 early EC, 67 EBL, and 67 HC was used to develop a predictive model for early EC diagnosis, and an independent validation set of 33 early EC, 33 EBL, and 33 HC was applied for validation of the model.

Materials & Methods

Study participants

We collected 300 retrospective serum samples from January 2021 to December 2022 from the Fujian Provincial Maternal and Child Health Hospital, including 100 early EC patients, 100 EBL patients, and 100 HCs. All study participants provided informed consent, and the study design was approved by the appropriate ethics review board (Grant No. 2023KYLLRD01058). The 100 early EC patients at TNM I/II stage were diagnosed by medically histopathology or cytology, and received no anti-cancer treatment, such as surgery, radiotherapy, or molecular targeted therapy. A total of 100 EBL patients were admitted to the Fujian Provincial Maternal and Child Health Hospital for primary treatment, including endometrial thickening, uterine fibroids, and endometriosis, during the same period. The 100 HCs were age-matched women receiving physical examinations at Fujian Provincial Maternal and Child Health Hospital, and showed no evidence of acute or chronic disease. All serum samples were collected in strict accordance with the following requirements: collection of five mL of peripheral blood from each subject prior to surgery; serum was separated at 3,000 rpm for 5 min, and frozen at −80 °C before use. This study was approved by the Institutional Review Board of Fujian Provincial Maternal and Child Health Hospital, and all enrolled participants provided written informed consent.

Immunohistochemistry

CST1 primary antibody (1:100, Abcam, Cambridge, UK) was added dropwise to the repaired and stained tissue sections and incubated overnight at 4 °C (Fuzhou Maixin Biotechnology Limited Company, Fuzhou, China). The tissues were then incubated with biotinylated secondary antibody and streptomyces anti-biotin protein-peroxidase solution, dried, and sealed by DAB color development and hematoxylin re-staining. Each section was randomly observed at five high-magnification fields and analyzed by two experienced pathologists. The staining intensity was graded as 0 (negative), 1 (weakly positive), 2 (moderately positive), or 3 (strongly positive).

Detection of serum CST1

The serum test kit for CST1 was the Human Cysteine Protease Inhibitor 1 Assay Kit (Jiangsu Enzyme-Free Industrial Co, Jiangsu, China). Serum CST1 was measured using enzyme-linked immunosorbent assay (ELISA), and the threshold value for serum CST1 in patients with early EC was determined by the highest discriminatory capacity (maximum sum of sensitivity and specificity) through the receiver operating characteristic (ROC) curve.

In addition, the performance of ELISA for the detection of serum CST1 was evaluated in terms of linearity, limit of detection, accuracy, precision, and resistance to interference.

For the detection limit and linearity, the CST1 calibrator (0 µg/L) was subjected to 20 intra-batch replicate determinations and the mean (X¯) and standard deviation (SD) of the 20 OD values measured were calculated, with X¯ ± 2 SD equal to the detection limit. After serial multiplicative dilutions, each dilution concentration of the CST1 calibrator was tested twice, and the mean OD was calculated. A linear regression curve of CST1 was plotted using the dilution concentration (C) as the vertical coordinate Y and the OD value as the horizontal coordinate X. The linear regression equation “C = a∗OD + b” was obtained, where a and b are the ELISA parameters indicating the conversion relationship between the OD value and CST1 level of the sample. The results demonstrated that the minimum detection limit for CST1 was 3.294 g/L (Table S1) and the correlation coefficient for CST1 in the range 0–1,600 µg/L was 0.993 (Fig. S1, Table S2).

To assess accuracy, the CST1 calibrators at concentrations of 100, 800, and 1,600 µg/L were mixed with certain concentrations of serum samples in a volume ratio of 1:9. The mixed samples were tested twice, the diluted samples were tested three times, and the mean level was calculated. The recoveries were calculated using the following formula: R=V0+V1×C−V1C1V0C0×100%

where V0 and C0 are the volume and level of calibrator, respectively; V1 and C1 are the volumes and levels of the serum samples, respectively; C is the concentration of the mixed samples; and R is the recovery rate. An 85%–115% recovery rate was acceptable. The results showed a recovery rate of 102.82% for CST1 (Table S3).

Precision was assessed according to the CLSI EP15-A2 standard. Samples with low and high CST1 levels were tested 20 times separately, and intra-batch CV < 10% was considered acceptable. CV values of 4.92% and 8.35% were determined for low and high CST1 samples, respectively (Table S4). Low and high CST1 samples were tested separately for five days, and each level was repeated three times to obtain mean values; inter-batch CV < 15% was considered acceptable. The results showed CV values of 7.63% and 9.27% for the low and high CST1 levels, respectively (Table S5).

Interference assessment was carried out according to the WS-T 416-2013 interference test guidelines: serum samples with CST1 levels of 48.5 and 87.61 µg/L were added to 2 g/L of hemoglobin, 342 µmol/L of bilirubin, and 37 mmol/L of triglycerides in a ratio of 1:9. Each sample was measured three times, and a relative deviation (δ) of <10% was considered acceptable. The results excluded that hemoglobin, bilirubin and triglycerides did not significantly interfere with CST1 determination (Table S6).

Detection of serum CA125 and HE4

Serum CA125 and HE4 levels were measured by chemiluminescent immunoassay kits on the Cobas 602 analyzer (Roche, Mannheim, Germany) according to the manufacturer’s instructions. The cutoff values for serum CA125 and HE4 levels in patients with early EC were determined based on the highest discriminatory capacity (maximum sum of sensitivity and specificity) using receiver operating characteristic (ROC) curves.

Statistical analysis

The SPSS software (version 25.0) was used for statistical analysis. One-way analysis of variance (ANOVA) was used for comparisons among three or more groups if the chi-square test was satisfied, and the Student–Newman–Keuls (SNK) method was chosen for multiple comparisons between groups. The Kruskal–Wallis test was used for non-parametric comparisons, and Kruskal–Wallis one-way ANOVA (K-sample comparison) was used for two-way comparisons. GraphPad Prism 9 was used to produce ROC curves, and the AUC values were calculated. P < 0.05 was considered statistically significant.

Results

Expression of CST1 protein in EC cancer tissues

Immunohistochemical results showed that the CST1 protein exhibited focal or diffuse distribution of tan granules in EC early-stage cancer tissues (Fig. 1A), whereas not expressed or weakly expressed in paired paracancerous tissues (Fig. 1B). Moreover, as shown in Table 1, CST1 was positive in 67.6% of CST1 protein in EC cancer tissues, which was significantly higher than that in paraneoplastic tissues (P < 0.01). The above studies indicated that CST1 was aberrantly highly expressed in the cancer tissues of patients with early EC.

Figure 1 Expression of CST1 protein in cancerous and paired paracancerous tissues of EC patients.

CST1 protein was strongly positive in cancerous tissues A (×100, ×400). CST1 protein was negative in paired paracancerous tissues B (×100, ×400).

Table 1 Expression of CST1 protein in cancerous tissues and paired paracancerous tissues of EC patients.

Characteristics	n	−	+	+ +	+ +  +	Positive rate	P	
Cancerous tissue	37	12	12	6	7	67.6% (25/37)	<0.01	
Paracancerous tissue	37	32	4	0	1	13.5% (5/37)		
TNM stage							<0.05	
I–II	27	11	8	4	4	59.3% (16/27)		
III–IV	10	1	4	2	3	90.0% (9/10)		
Notes.

EC, Endometrial cancer; TNM, Tumor node metastasis.

Levels of serum biomarkers in the training set

As shown in Table 2 and Figs. 2A–2C, the levels of serum CST1 and HE4 were significantly higher in the early EC group than in the EBL/HC group (P < 0.05). However, there was no significant difference in serum CA125 levels between the early EC and EBL/HC groups (P > 0.05).

Table 2 Comparison of the diagnostic performances of serum CST1, CA125, and HE4 for Early EC patients in the training set.

Variable	Early EC (n = 67)	EBL (n = 67)	HC (n = 67)	P-value	
Age (Year)	52.5 ± 7.3	47.6 ± 6.8	52.3 ± 8.7	>0.05	
CST1 (µg/L)	406.3 (274.5–440.65)	286.84 (225.0–312.67)	274.30 (215.5–311.9)	<0.01	
CA125 (U/mL)	28.15 (11.4–29.15)	27.04 (14.09–35.6)	14.13 (8.15–16.55)	>0.05	
HE4 (pmol/L)	74.86 (35.98–77.20)	40.23 (31.46–47.99)	35.57 (31.5–43.65)	<0.05	
Notes.

EC, Endometrial cancer; EBL, Endometrial benign lesion; HC, Health control.

Figure 2 Performances of serum CST1, CA125, and HE4 for early EC patients in the training set.

Boxplot, scatter of serum CST1 (A), CA125 (B), HE4 (C), and the ROC curves of serum CST1, CA125, and HE4, and combinations of indicators (D). An asterisk (*) indicates P <  0.05; two asterisks (**) indicate P <  0.01.

Diagnostic performance of serum biomarkers in the training set

As shown in Table 3 and Fig. 2D, serum CST1 had an AUC of 0.715, sensitivity of 31.3%, and specificity of 90.3% for patients with early EC; serum CA125 and HE4 had AUCs of 0.577 and 0.706, sensitivity of 20.9% and 23.9%, and specificity of 88.1% and 95.5%, respectively, for patients with early EC. Among all possible combinations, the combination of serum CST1 and HE4 had the best diagnostic performance for early EC, with a maximum AUC of 0.788 (95% CI [0.720–0.856]), sensitivity of 49.3%, specificity of 92.5%, maximum positive predictive value of 76.7%, and negative predictive value of 78.5%.

Table 3 Comparison of the diagnostic performances of serum CST1, CA125, and HE4 for Early EC patients in the training set.

Marker	AUC	SE	95% CI	Sensitivity (%)	Specificity (%)	PPV (%)	NPV (%)	P-value	
CST1	0.715	0.034	0.647–0.781	31.3	90.3	61.8	72.5	<0.01	
CA125	0.577	0.043	0.492–0.662	20.9	88.1	46.7	69.0	>0.05	
HE4	0.706	0.041	0.626- 0.786	23.9	95.5	72.7	71.5	<0.05	
CST1+CA125	0.753	0.034	0.685–0.821	28.4	93.3	67.9	72.3	0.009	
CST1+HE4	0.788	0.034	0.720–0.856	49.3	92.5	76.7	78.5	<0.001	
CA125+HE4	0.668	0.039	0.592–0.744	34.3	91.0	65.7	73.5	<0.001	
CST1+CA125+HE4	0.679	0.038	0.605–0.753	38.9	91.8	70.3	69.9	<0.001	
Notes.

AUC areas under the curve

SE standard error

CI confidence interval

PPV positive predictive value

NPV negative predictive value

Serum biomarker levels in the validation set

Table 4 and Figs. 3A, 3B showed that the levels of serum CST1 and HE4 expression were significantly higher in the early EC group than in the EBL/HC groups (P < 0.05). However, the difference in serum CA125 levels was not significant.

Table 4 Clinical data for Early EC, EBL and HC groups in the validation set.

Variable	Early EC (n = 33)	EBL (n = 33)	HC (n = 33)	P-value	
Age (Year)	54.9 ± 8.7	49.2 ± 5.4	55.1 ± 10.7	>0.05	
CST1 (µg/L)	373.44 (278.5–431.05)	304.18 (245.0–325.65)	298.89 (233.0–344.95)	<0.01	
HE4 (pmol/L)	66.45 (42.47–73.59)	39.52 (28.94–48.38)	35.03 (29.49–41.62)	<0.05	
Notes.

EC Endometrial cancer

EBL Endometrial benign lesion

HC Health control

Figure 3 Performances of serum CST1 and HE4 for early EC patients in the validation set.

Boxplot, scatter of serum CST1 (A), HE4 (B), and the ROC curves of serum CST1, HE4, and CST1+HE4 (C). An asterisk (*) indicates P <  0.05; two asterisks (**) indicate P <  0.01.

Diagnostic performance of serum biomarkers in the validation set

The results of the validation set demonstrated that serum CST1 had an AUC of 0.719, with a sensitivity of 39.4% and specificity of 86.4%; serum HE4 had an AUC of 0.686, with a sensitivity of 24.3% and specificity of 96.9%; the combination of serum CST1 and HE4 had an AUC of 0.824, with a sensitivity of 48.5% and specificity of 92.4% (Table 5, Fig. 3C).

Table 5 Comparison of the diagnostic performances of serum CST1 and HE4 for Early EC patients in the validation set.

Marker	AUC	SE	95% CI	Sensitivity (%)	Specificity (%)	PPV (%)	NPV (%)	P-value	
CST1	0.719	0.058	0.606–0.831	39.4	86.4	62.5	70.1	<0.01	
HE4	0.686	0.056	0.576–0.796	24.3	96.9	80.0	71.9	0.015	
CST1+HE4	0.824	0.043	0.740–0.907	48.5	92.4	76.2	78.2	<0.001	
Notes.

AUC areas under the curve

SE standard error

CI confidence interval

PPV positive predictive value

NPV negative predictive value

Discussion

The correlation between aberrant CST1 expression and the diagnosis, disease assessment, and prognosis of some malignancies has attracted increasing attention in recent years (Jiang et al., 2015; Wang et al., 2021; Lai et al., 2022). CST1 is a secreted protein belonging to the second subfamily of the CST superfamily, and includes seven exocrine proteins from cystatin SN (CST1) to cystatin F (CST7). Interestingly, CST1 presents a specific distribution in body fluids and tissues, with limited expression in seminal fluid, lacrimal fluid, gallbladder fluid, submandibular gland, lacrimal gland, and gallbladder (Abrahamson et al., 1986; Barka et al., 1991; Dickinson et al., 1993). A large body of evidence indicates the existence of aberrantly high ectopic expression of CST1 existed in cancerous tissues of some malignancies (Dai et al., 2017; Wang et al., 2021; Zhang et al., 2023), suggesting that CST1 might be present in the sera of patients at early stages of EC. Based on observations of high CST1 expression in both cancerous tissues and sera of patient with early EC by a small cohort pilot study, it is reasonable to deduce that serum CST1 might serve as a promising serological biomarker for early diagnosis of patients with EC. Encouragingly, in this study, by detecting 201 serum samples of a training set comprised of 67 early EC patients, 67 EBL patients, and 67 HCs, we achieved the expected results, evidenced by the fact that serum CST1 in the early EC group was significantly higher than that in EBL/HC groups (P < 0.05), and had an AUC of 0.715 superior to that of HE4 (0.706) and CA125 (0.577), with 31.3% sensitivity at 90.3% specificity. Additionally, the diagnostic value of serum CST1 for early EC was well validated by an independent validation set, comprised of 33 early EC patients, 33 EBL patients, and 33 HCs.

Considering the relatively limited sensitivity of individual detection, we further performed the evaluation of the diagnostic performances of all possible combinations of serum CST1 with traditional tumor markers, CA125 and HE4, for patients with early EC. Some studies have shown that HE4 correlates better with EC than CA125 and is of interest in the diagnosis, prognosis and recurrence monitoring of EC (Li et al., 2009; Behrouzi, Barr & Crosbie, 2021). Our study found that the combination of serum CST1 and HE4 had the optimal diagnostic performance for early EC patients, with an AUC of up to 0.788, and sensitivity of 49.3% at a guaranteed specificity of 92.5%. Moreover, this combination also exhibited sufficient diagnostic potential for early EC in an independent validation set, with an AUC of 0.824, and 48.5% sensitivity at 92.4% specificity.

In summary, we verified that the ELISA method has good detection performance for serum CST1, and clarified the diagnostic value of serum CST1 for early EC, while the combined detection of CST1 and HE4 can further improve the early diagnostic efficacy of EC patients. In addition, Since this project is a single center research and has certain limitations, it can further verify the diagnostic value of serum CST1 through joint research by multiple centers in the later period.

Supplemental Information

Supplemental Information 1 Performance of serum CST1 for early EC patients in a small sample cohort

Boxplot, scatter of serum CST1 (A), and the ROC curve of serum CST1(B).

Click here for additional data file.

Supplemental Information 2 Supplementary Tables

Click here for additional data file.

Supplemental Information 3 Raw data

Click here for additional data file.

Supplemental Information 4 Raw data for early stage endometrial cancer

Click here for additional data file.

Supplemental Information 5 Raw data for advanced endometrial cancer

Click here for additional data file.

Supplemental Information 6 Raw data statistics of benign diseases

Click here for additional data file.

Supplemental Information 7 Raw data from a healthy control population

Click here for additional data file.

Supplemental Information 8 Raw data of EC pathological tissue

Click here for additional data file.

We are grateful to all the participants of the present study and the Oncology Laboratory of Fujian Maternal and Child Health Hospital for providing support and assistance.

Additional Information and Declarations

Competing Interests

Author Contributions

Human Ethics

Data Availability

The authors declare there are no competing interests.

Wenhui Zhong conceived and designed the experiments, performed the experiments, analyzed the data, prepared figures and/or tables, authored or reviewed drafts of the article, and approved the final draft.

Yunliang Liu conceived and designed the experiments, performed the experiments, analyzed the data, prepared figures and/or tables, authored or reviewed drafts of the article, and approved the final draft.

Liangming Zhang performed the experiments, analyzed the data, prepared figures and/or tables, and approved the final draft.

Wanzhen Zhuang performed the experiments, analyzed the data, prepared figures and/or tables, and approved the final draft.

Jianlin Chen performed the experiments, analyzed the data, prepared figures and/or tables, and approved the final draft.

Zhixin Huang performed the experiments, analyzed the data, prepared figures and/or tables, and approved the final draft.

Yue Zheng performed the experiments, analyzed the data, prepared figures and/or tables, and approved the final draft.

Yi Huang conceived and designed the experiments, authored or reviewed drafts of the article, and approved the final draft.

The following information was supplied relating to ethical approvals (i.e., approving body and any reference numbers):

Ethics Committee of Fujian Provincial Maternal and Child Health Hospital

The following information was supplied regarding data availability:

The raw data is available in the Supplementary Files.

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
