# Peer review of "Combination of serum CST1 and HE4 for early diagnosis of endometrial cancer"

_PeerJ, doi:10.7717/peerj.16424_

## Round 0.1 · original submission · Major Revisions

Please address the concerns of all reviewers and amend the manuscript accordingly.

**Language Note:** PeerJ staff have identified that the English language needs to be improved. When you prepare your next revision, please either (i) have a colleague who is proficient in English and familiar with the subject matter review your manuscript, or (ii) contact a professional editing service to review your manuscript. PeerJ can provide language editing services - you can contact us at [email protected] for pricing (be sure to provide your manuscript number and title). – PeerJ Staff

·

Basic reporting

In the article by Zhong et al., " Combination of serum CST1 and HE4 for early diagnosis of endometrial cancer", they aimed to identify different serological biomarkers for early diagnosis of endometrial cancer. The authors have conducted ELISA and chemiluminescent immunoassays to detect the presence of CST1, HE4, and CA125 in serum. Their findings suggest that CST1 can be a prospective serological biomarker for early EC diagnosis, and the combination of CST1 and HE4 will contribute further to the diagnosis of patients with early-stage EC.

Below, I provide comments to improve this article.

1. Figure 1: Please label the image samples. Indicate the tissue of interest with an arrow. Provide a scale bar for images. Please provide zoomed-out images of whole tissue sections.
2. Figure 2+3: Please refer clearly to the statistical analysis done for each image. The P value should be included either in the main figure or in the figure legends. Please mention the sample size and number of biological replicates.
3. Please mention the full form of any abbreviation used for the first time in the text. EBL and HC?
4. Authors should provide TMA (tissue microarrays) to establish the expression level of the CST1 protein in cancer vs normal tissue. Expression of the CST1 protein at different stages of tumor progression would be useful.
5. Does the CST1 protein level vary based on different treatments in patients?
6. Figure 1: How many patient samples did you check? Please provide IHC image analysis.
7. Do you have patient or participant information from all three groups? Such as treatment history, ethnicity, stage of tumors, histological subtypes, etc. It would be useful information to decide the relevance of the use of these markers for diagnostic purposes.

Experimental design

1. Authors should provide TMA (tissue microarrays) to establish the expression level of the CST1 protein in cancer vs normal tissue. Expression of the CST1 protein at different stages of tumor progression would be useful.
2. Does the CST1 protein level vary based on different treatments?
3. Figure 1: How many patient samples did you check? Please provide IHC image analysis.
4. Do you have patient or participant information from all three groups? Such as treatment history, ethnicity, stage of tumors, histological subtypes, etc. It would be useful information to decide the relevance of the use of these markers for diagnostic purposes.

Validity of the findings

1. Figure 1: Please label the image samples. Indicate the tissue of interest with an arrow. Provide a scale bar for images. Please provide zoomed-out images of whole tissue sections.
2. Figure 2+3: Please refer clearly to the statistical analysis done for each image. The P value should be included either in the main figure or in the figure legends. Please mention the sample size and number of biological replicates.

Reviewer 2 ·

Basic reporting

The manuscript submitted by Wenhui Zhong et al. studies the detection of CST1 (cystatin SN) and HE4 (human epididymis protein 4) as an indicator for the early diagnosis of endometrial cancer. Three makers, CST1, HE4, and CA125 (glycated antigen 125) were evaluated by ELISA and chemiluminescent immunoassay in the serum samples. A training data set was developed to create a predictive model for early EC (endometrial cancer) diagnosis using early EC, EBL (endometrial benign lesion), and HC (health control) patients’ samples, which was validated by an independent validation data set. The manuscript is clearly written in professional, unambiguous language.

Experimental design

1. The abstract, results & discussion are clearly written with well-designed experiments and analysis. However, the manuscript has a number of major and minor issues that should be addressed before acceptance.

2. The experiments are designed well, and the correct statistical analysis was performed.

3. In Figure 1, please include the scale bar and use arrows to mark the CST1 protein expression, to make it easy for the readers.

Validity of the findings

Detection of CST1 (cystatin SN) and HE4 (human epididymis protein 4) was shown as an indicator for the early diagnosis of endometrial cancer. An independent validation data set was used to validate the obtained results.

Additional comments

1. Please use the full abbreviation of the words when it is used for the first time in the manuscript. Example, line 36, CST1; line 38, EC, line 91, TNM, etc.

2. Line 71-74, For example, glycated antigen 125 (CA125) and …. renal function, respectively. Could you comment on the CST1 as well? Does CST1 have a dependence on the female physiological cycle, age, and renal function?

3. Line 128, section “detection of serum CST1”, please cite the paper reference for the method used for the calculations.

4. Line 191, please elaborate on the 67.7% of CST1 expression from the EC cancer tissues. How the percentage was calculated?

5. Line 222, please include the reference “The correlation between aberrant CST1 expression and the diagnosis, …. in recent years.”

6. I would recommend the authors to include the following two references in the discussion and elaborate the discussion part. Please gel together the known information about the CST1 and HE4. Were these markers used for the other types of cancer? How are the obtained results of the HE4 marker different from the data obtained in the Li J et al. (2009) Expert Rev Mol Diagn? Please comment on this.

a. Cite the paper by Li J, et al. HE4 as a biomarker for ovarian and endometrial cancer management. Expert Rev Mol Diagn. 2009 Sep;9(6):555-66.

b. Lai Y, et al. Identification and Validation of Serum CST1 as a Diagnostic Marker for Differentiating Early-Stage Non-Small Cell Lung Cancer from Pulmonary Benign Nodules. Cancer Control. 2022 Jan-Dec;29:10732748221104661.

7. Could not open the data files named, Data from a healthy control population, Data for early-stage endometrial cancer, and Data for advanced endometrial cancer. Could you attach the file in Microsoft Excel worksheet format?

·

Basic reporting

no comment

Experimental design

no comment

Validity of the findings

no comment

Additional comments

Zhong et. al in this manuscript shows that serum CST1 combined with 37 glycated antigen 125 (CA125) and human epididymis protein 4 (HE4) are prospective serological biomarkers for early EC diagnosis to improve the diagnosis in the future. However, there are two points needed to be revised:
1. Please use full name instead of abbreviation when it first appears. such as Line36 CTS1, Line47 AUC, etc. It is necessary to show the complete name in the note when you use the abbreviation in table. such as table 4.
2. The authors address that “combinations” of biomarkers work better. Please provide a comparison of the diagnostic effectiveness of each component in combination versus alone. Try to give more detailed explanation between the relationship of the combined components in diagnosis. The specificity of early EC biomarkers is important in diagnosis. Please discuss that to make conclusion solid.

---

## Round 0.2 · accepted · Accept

All issues were adequately addressed and revised manuscript is acceptable now.

·

Basic reporting

The authors addressed the reviewer's concerns in full. Additional supporting data were included, and the text was revised with greater clarity, which has strengthened the manuscript considerably.

Experimental design

NA

Validity of the findings

NA

Additional comments

NA

Reviewer 2 ·

Basic reporting

All my concerns have been addressed and the manuscript has been revised accordingly. I recommend the publication of the revised manuscript in PeerJ.
One very minor suggestion, Line 245, please change the word “some” with “previous” in the sentence “Some studies have shown that HE4 correlates better with EC than CA125 and is of interest in the diagnosis, prognosis, and recurrence monitoring of EC”.

Experimental design

N/A

Validity of the findings

N/A

Additional comments

N/A